# Evaluation of two automated low-cost RNA extraction protocols for SARS-CoV-2 detection

**Fernando Lázaro-Perona**[☉], **Carlos Rodriguez-Antolín**[iD][☉], **Marina Alguacil-Guillén**[iD], **Almudena Gutiérrez-Arroyo, Jesús Mingorance**[iD]*, **Julio García-Rodriguez, on behalf of the SARS-CoV-2 Working Group**[¶]

Servicio de Microbiología, Hospital Universitario La Paz, IdiPAZ, Madrid, Spain

☉ These authors contributed equally to this work.
¶ Membership of the SARS-CoV-2 Working Group is provided in the Acknowledgments.
* jesus.mingorance@idipaz.es

**Data Availability Statement:** All relevant data are within the paper and its Supporting Information files.

## Abstract

### Background

Two automatable in-house protocols for high-troughput RNA extraction from nasopharyngeal swabs for SARS-CoV-2 detection have been evaluated.

### Methods

One hundred forty one SARS-CoV-2 positive samples were collected during a period of 10-days. In-house protocols were based on extraction with magnetic beads and designed to be used with either the Opentrons OT-2 (OT-2$_{in-house}$) liquid handling robot or the MagMAX$^{TM}$ Express-96 system (MM$_{in-house}$). Both protocols were tested in parallel with a commercial kit that uses the MagMAX$^{TM}$ system (MM$_{kit}$). Nucleic acid extraction efficiencies were calculated from a SARS-CoV-2 DNA positive control.

### Results

No significant differences were found between both in-house protocols and the commercial kit in their performance to detect positive samples. The MM$_{kit}$ was the most efficient although the MM$_{in-house}$ presented, in average, lower Cts than the other two. In-house protocols allowed to save between 350€ and 400€ for every 96 extracted samples compared to the commercial kit.

### Conclusion

The protocols described harness the use of easily available reagents and an open-source liquid handling system and are suitable for SARS-CoV-2 detection in high throughput facilities.

**Funding:** The author(s) received no specific funding for this work.

**Competing interests:** The authors have declared that no competing interests exist.

## Introduction

The SARS-CoV-2 pandemic has called for the use of massive qPCR tests in order to detect positive cases and to trace contacts to stop community transmission. Prior to qPCR testing, clinical samples are usually subject to RNA extraction [1–4]. Given the number of samples tested every day, manual RNA extraction methods are unfeasible for most facilities and thus, automatic systems are widely used for this task [5–7]. As a drawback, automatic systems significantly increase the final costs, which can hinder massive testing in some areas. Moreover, due to the increased demand, stock shortage of extraction reagents has caused major delays in diagnostics.

In this work, two low-cost alternatives for automatic RNA extraction are described. The first one using the OT-2 system (Opentrons, New York, NY, USA), an open-source liquid handler robot capable of automating self-designed protocols, and the second using the rapid and easy-to-use nucleic acid extractor MagMAX$^{TM}$ Express-96 system (Thermo Fisher Scientific, Waltham, MA, USA). The later extracts up to 96 samples in 30 min, though it requires a previous manual dispensation of the reagents, magnetic beads and samples in 96-well plates adding 30 min. As an alternative, the OT-2$_{in-house}$ protocol can process up to 48 samples in 104 min in a fully automated manner.

## Methods

### Sample collection

During ten days 141 consecutive SARS-CoV-2 positive nasopharyngeal swabs with viral transport medium (Deltalab, Barcelona, Spain) that had been stored at 4°C were collected. Before processing, the samples were inactivated by mixing 500 μL of the viral medium and 500 μL of 4M guanidine Isothiocyanate (GTC) (Qiagen, Hilden, Germany) with 5 μg/mL carrier RNA, followed by heating the samples 2 min at 80°C and a short vortex mixing.

### Equipment and reagents

Automatic extraction of nucleic acids was done using two systems: the MagMAX$^{TM}$ Express-96 Deep Well Magnetic Particle Processor (King Fisher Instrument, Thermo Fisher Scientific, Waltham, MA, USA) and the open system OT-2 (Opentrons, New York, NY, USA) with a GEN1 magnetic module (Opentrons, New York, NY, USA) and an in-house protocol. Mag-MAX™ Express 96 plates and Deep Well plates (Thermo Fisher Scientific, Waltham, MA, USA) were used with the two systems.

Nucleic acid extraction was done with three methods: 1) Using the MagMAX$^{TM}$ with the commercial MagMAX CORE Nucleic Acid Purification kit (MM$_{kit}$) (Thermo Fisher Scientific, Waltham, MA, USA) according to the manufacturer instructions; 2) The OT-2 system with generic reagents (OT-2$_{in-house}$) such as ethanol (Emsure®, Merck KGaA, Darmstadt, Germany), 2-Propanol (Emsure®, Merck KGaA, Darmstadt, Germany), Elution Buffer (Omega BIO-TEK, Norcross, GA, USA), Nuclease free water (Ambion$^{TM}$, Thermo Fisher Scientific, Waltham, MA, USA) and magnetic beads (Mag-Bind® TotalPure NGS, Omega Bio-Tek, Norcross, GA, USA); 3) MagMAX$^{TM}$ with the same protocol as the commercial kit but the reagents used in the OT-2 method (MM$_{in-house}$). In-house protocols are a modification of the procedure described by Hui He et al. [8]. Briefly, inactivated respiratory samples are mixed in a 1:1 proportion with isopropanol to a final volume of 500 μl for the OT-2$_{in-house}$ and 1000 μL for the MM$_{in-house}$, 40 μL of magnetic beads are added and the mixtures are incubated for 5 min at room temperature. Next, magnetic beads are pulled to one side of the tubes with a magnet and supernatant is discarded. Magnetic beads are then washed two times with 500 μL of freshly prepared ethanol 70%. After the second wash, the ethanol 70% is discarded and the

**Table 1. Steps, reagents and volumes (µL) used in the three protocols evaluated.**

| | Step1 (Reagent mix) | | | | Step2 (Wash) | Step3 (Wash) | Step4 (Elution) |
|---|---|---|---|---|---|---|---|
| | Sample | Lysis/Binding buffer | Isopropanol absolute | Magnetic Beads | Wash 1 buffer* | Wash 2 buffer* | Elution buffer |
| MM$_{kit}$ | 200 | 700 | - | 30 | 500 | 500 | 90 |
| OT-2$_{in-house}$ | 250 | - | 250 | 40 | 500 | 500 | 100 |
| MM$_{in-house}$ | 500 | - | 500 | 40 | 500 | 500 | 90 |

*The MM$_{in-house}$ and OT-2$_{in-house}$ protocols use ethanol 70% in the two washing steps.

magnetic beads are air dried at room temperature. Finally, the beads are resuspended in 100 µL of the elution buffer and separated with the magnet again to recover the eluted viral RNA (Table 1).

## Protocol design and validation

MM$_{kit}$ protocol:

(Taken from https://assets.thermofisher.com/TFS-Assets/LSG/manuals/MAN0015944_MagMAXCORE_NA_Kit_UG.pdf)

1. Prepare two Deep well plates, one with 500µL of MagMAX™ CORE Wash Solution 1 per well, and another one with 500µL of MagMAX™ CORE Wash Solution 2 per well. Prepare a MagMAX™ Express 96 microtiter plate with 90µL of MagMAX™ CORE Elution Buffer per well.

2. Prepare a mixture of 20µL of MagMAX™ CORE Magnetic Beads and 10µL of MagMAX™ CORE Proteinase K per sample (beads/PK mix).

3. Prepare a mixture of 300µL MagMAX™ CORE Lysis Solution and 300µL MagMAX™ CORE Binding Solution per sample (Lysis/Binding solution).

4. Dispense in a Deep well plate 30µL of beads/PK mix, 700µL of Lysis/Binding solution and 200µL of sample per well.

5. Put all the plates in the system and run the script MagMAX_CORE_KF-96.

MM$_{in-house}$ protocol:

1. Prepare two Deep well plates with 500µL of Ethanol 70% per well. Prepare a MagMAX™ Express 96 microtiter plate with 90µL of Elution Buffer (Omega BIO-TEK, Norcross, GA, USA) per well.

2. Dispense in a Deep well plate 40µL of magnetic beads (Mag-Bind® TotalPure NGS, Omega Bio-Tek, Norcross, GA, USA), 500µL of Isopropanol and 500µL of sample per well.

3. Put all the plates in the system and run the script MagMAX_CORE_KF-96.

OT-2$_{in-house}$protocol:

1. Dispense in a Deep well plate 40µL of magnetic beads (Mag-Bind® TotalPure NGS, Omega Bio-Tek, Norcross, GA, USA), 250µL of Isopropanol and 250µL of sample per well. Mix by pipetting five times and incubate 5 min at room temperature.

2. Activate the GEN1 magnetic module 4 min.

3. Collect the supernatant and discard.

4. Add 500μL of Ethanol 70%, collect and discard.

5. Add 500μL of Ethanol 70%, collect and discard.

6. Air dry for 4 min.

7. Turn off the GEN1 magnetic module and add 100μL of Elution Buffer (Omega BIO-TEK, Norcross, GA, USA).

8. After 30 sec turn on the GEN1 magnetic module.

9. After 90 sec collect the supernatant and transfer to a 96-well microtiter plate.

## The sample input was the maximum volume allowed by the automatic pipetting systems and the volumes of the other reagents and therefore was different for the three methods

OT-2$_{in-house}$ protocol was written in Python according to the Opentrons instructions. The scripts have been deposited in the GitHub repository (https://github.com/HULPopentrons/RNA_extraction_OT2opentrons).

To validate the performance of the in-house protocols for nucleic acid extraction, simulated samples were made with standard, low and very low viral loads using the DNA positive control TaqMan 2019-nCoV Control Kit v1 (Thermo Fisher Scientific, Waltham, MA, USA). The Control Kit comes at a concentration of 1 x 10$^4$ copies/μL. The standard viral load sample was prepared by mixing 10 μL of the positive control with 490 μL of viral transport medium and 500 μL of GTC. The low and very-low viral loads were prepared in the same manner but using ten-fold serial dilutions of the positive control. All mock samples were prepared in triplicate and processed in parallel with the OT-2$_{in-house}$, the MM$_{kit}$ and the MM$_{in-house}$ and then tested by qPCR with the TaqMan 2019-nCoV Assay Kit v1. Final concentration of the positive control in the simulated samples with the standard, low and very-low viral loads was 1 x 10$^6$ copies/mL, 1 x 10$^5$ copies/mL and 1 x 10$^4$ copies/mL respectively. Negative controls were included in all the runs.

Nucleic acid extraction efficiency was calculated by comparing the Ct values of the positive controls in the simulated samples (Ct$_{ss}$) with the Ct values of the positive controls prepared directly from the stock correcting the amounts to match the dilution factor and the amount of initial sample used in each protocol (Ct$_{pc}$) (R = 2$^{-\Delta Ct}$ = 2$^{-(Ctss-Ctpc)}$).

## qPCR

Nucleic acid amplification of the SARS-CoV-2 viral RNA was done using the TaqMan 2019-nCoV Assay Kit v1 that targets the *orf1ab*, spike (S), nucleocapsid (N) and human RNaseP genes and the TaqMan 2019-nCoV Control Kit v1 as positive control with the qPCR conditions recommended by the manufacturer (https://assets.thermofisher.com/TFS-Assets/LSG/manuals/MAN0019096_TaqMan2019nCoVAssayKit_PI.pdf). All qPCR assays were performed in a CFX96 Touch Real-Time PCR Detection System (Bio-Rad, Hercules, CA, USA). To reduce inter-assay variability, extracted nucleic acid samples were dispensed automatically in qPCR strips with another OT-2 module.

## Statistical analyses

The three protocols were compared pairwise. McNemar's test was used to compare their performance in assigning samples as positive or negative. Ct values are not distributed normally, therefore the Wilcoxon Signed-Rank test was used for comparing the Ct values of each target.

For each target only the samples that were amplified by the three methods were considered. Median confidence intervals were calculated with a bootstrap method.

All the statistical tests were performed with the IBM SPSS Statistics 24.0.0.0 package (SPSS Inc., Chicago, IL, USA).

## Results

### Efficiencies of the extraction protocols

Nucleic acid extraction efficiencies of the two in-house protocols were compared with the $MM_{kit}$ protocol by extracting the mock samples prepared with the SARS-CoV-2 positive control that simulate different viral loads. The $MM_{kit}$ and $OT-2_{in-house}$ protocol successfully amplified the *orf1ab*, S and N targets in all mock samples with the standard viral load. For these samples, the mean extraction efficiency of all replicates and genes was 33.9% (SD: 13.8) for the $MM_{kit}$, 19.6% (SD: 2.2) for the $OT-2_{in-house}$ protocol and 16.0% (SD: 5.03) for the $MM_{in-house}$.

In the low viral load mock samples, the $MM_{kit}$ failed to amplify the S target in all samples and the N target in one of the replicates while the $OT-2_{in-house}$ and $MM_{in-house}$ detected all the genes. Finally, the $MM_{kit}$ and $OT-2_{in-house}$ protocols failed to amplify the very-low viral load samples, except for one replicate in which the N gene could be amplified with the $OT-2_{in-house}$ protocol but the $MM_{in-house}$ could detect the positive control in all samples, one of them with the three targets, one with them with the *orf1ab* and N targets and the last one with the *orf1ab* target (S1 Table).

### Nucleic acid extraction from clinical respiratory samples

The 141 positive clinical samples collected were subjected to nucleic acid extraction in parallel with the $MM_{kit}$, the $OT-2_{in-house}$, and the $MM_{in-house}$ methods and the eluted SARS-CoV-2 RNA was tested by qPCR. Positive/negative results and amplification cycle threshold values (Ct) for each target were registered. According to the manufacturer's instructions samples were considered positive when at least one of the three-targeted regions amplified with a Ct<40. Negative controls included in all the runs yielded in all cases negative PCR results.

Of the 141 samples, 123 were positive for SARS-CoV-2 by at least one extraction method while 18 were negative by all three methods. This was probably due to sample degradation during the collection period, because they had been stored for about 48 h at 4°C [9]. By method, 114 samples tested positive using the $MM_{kit}$, 111 with the $OT-2_{in-house}$ and 118 with the $MM_{in-house}$. Pairwise comparisons found no significant differences in their performances for detecting SARS-CoV-2 ($MM_{kit}$ Vs $OT-2_{in-house}$, $P = 0.5465$; $MM_{kit}$ Vs $MM_{in-house}$, $P = 0.3865$; $OT-2_{in-house}$ Vs $MM_{in-house}$, $P = 0.0961$). Eighteen samples were negative by some of the three methods. These had high Ct values, with median values of the *orf1ab* and N targets of 38.68 and 38.19 respectively (in those samples the S target was not amplified by any method). Pairwise linear regression and correlation analyses comparing all methods and targets showed $R^2$ values between 0.85 and 0.95 and Bland-Altman analyses showed the agreement to be uniform through the whole Ct range (S1 Fig).

The three targets (*orf1ab*, N and S) were detected in 43% of the samples extracted with $MM_{kit}$, 49% of those extracted with $OT-2_{in-house}$ and 49% with $MM_{in-house}$ (Fig 1). In 34%, 39% and 30% the *orf1ab* and N targets were detected, and in 19%, 10% and 17% only the N target was amplified. Few samples were considered positive by the amplification of either the *orf1ab* target alone (6), the *orf1ab* + S (1), or the N + S (5) targets, and no sample had the S gene as the only positive marker. In fact, the S target in this assay had an evident lack of sensitivity and would be irrelevant to take diagnostic decisions. Samples with two or three targets detected

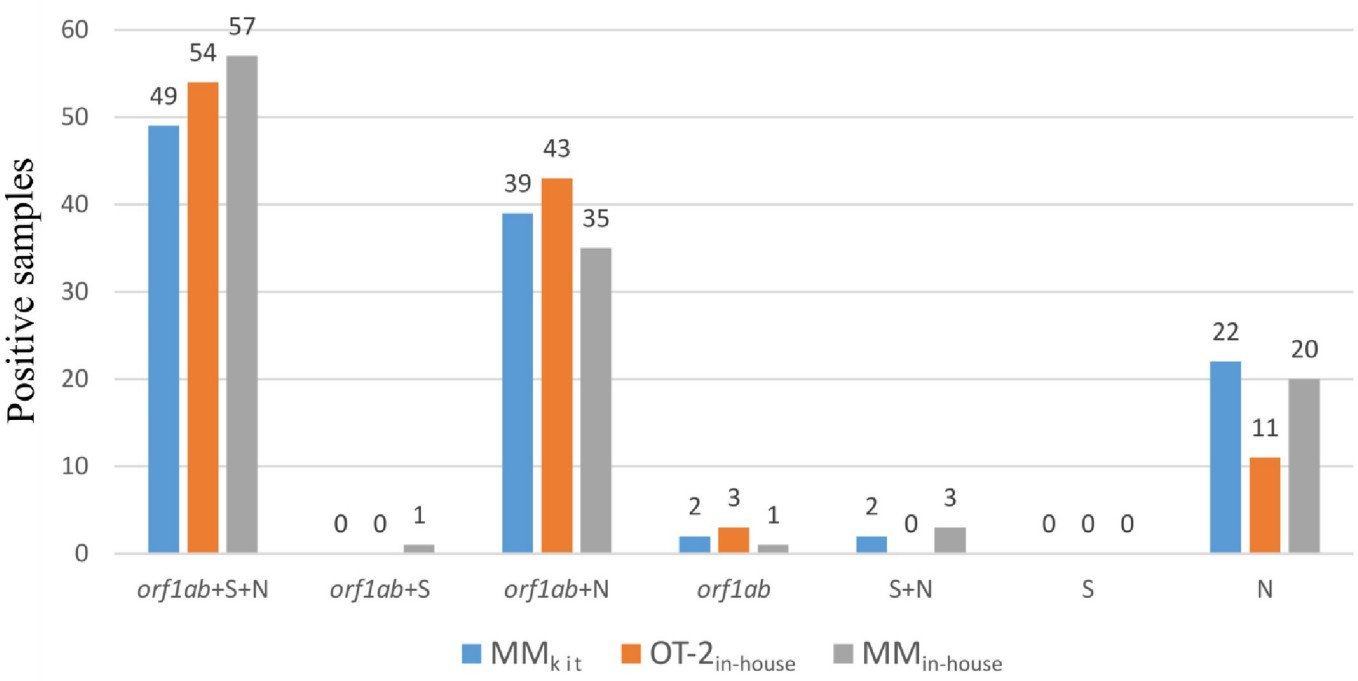

**Fig 1. Percentages of SARS-CoV-2 samples positive for the *orf1ab*, S and N targets with the MM$_{kit}$, OT-2$_{in-house}$ and MM$_{in-house}$ methods.** The numbers over the bars indicate the percentage of positive samples for each target. Overall, 114 (81%) samples were positive with MM$_{kit}$, 111 (79%) with OT-2$_{in-house}$ and 118 (84%) with MM$_{in-house}$.

were 87% (97 samples) with the OT-2$_{in-house}$ protocol, 82% with the MM$_{in-house}$ (95 samples) and 79% with the MM$_{kit}$ (90 samples).

When considering paired samples, no significant differences were found in the Cts between the MM$_{kit}$ and the OT-2$_{in-house}$ protocol in the Ct values for the *orf1ab* ($P = 0.437$), N ($P = 0.686$) and S ($P = 0.794$) targets (Fig 2). The MM$_{in-house}$ protocol did present significantly lower Cts in all targets compared to the MM$_{kit}$ (*orf1ab*; $P < 0.00001$, N; $P < 0.00001$, S; $P = 0.00148$) and the OT-2$_{in-house}$ (*orf1ab*; $P < 0.00001$, N; $P = 0.00008$, S; $P = 0.00252$).

The median amplification cycle (CI:95%) of the *orf1ab* target using the MM$_{kit}$, the OT-2$_{in-house}$ and the MM$_{in-house}$ were, respectively, 35.53 (33.82–36.36), 35.58 (34.33–36.21) and 34.8 (33.71–35.29). For the S gene 30.16 (28.56–33.08), 31.32 (29.22–32.88) and 31.07(29.36–32.90) and the N target 34.83 (33.93–35.97), 34.64 (33.42–35.29) and 34.28 (33.52–35.10).

## Extraction costs and hands-on time

Nucleic acid extraction of 96 samples using the OT-2$_{in-house}$ protocol had an overall cost of 107 € (37€ reagents and 70€ labware) with a hands-on time of 10 min and an extraction time of 3hours and a half (the script handles 48 samples in each run so it has to be completed two times). The MM$_{in-house}$ cost was 66€ (37€ reagents and 29€ labware) and the MM$_{kit,}$ 472€ (443 € reagents and 29€ labware). Both protocols had a hands-on time of 40 min and an extraction time of 30 min. Notably, although the OT-2 protocol for 96 samples was 40€ more expensive than the MM$_{in-house}$, the required initial investment is significantly lower, being approximately 10,000€ for the OT-2 system (including modules and pipettes) and 50,000€ for the Mag-MAX$^{TM}$ system.

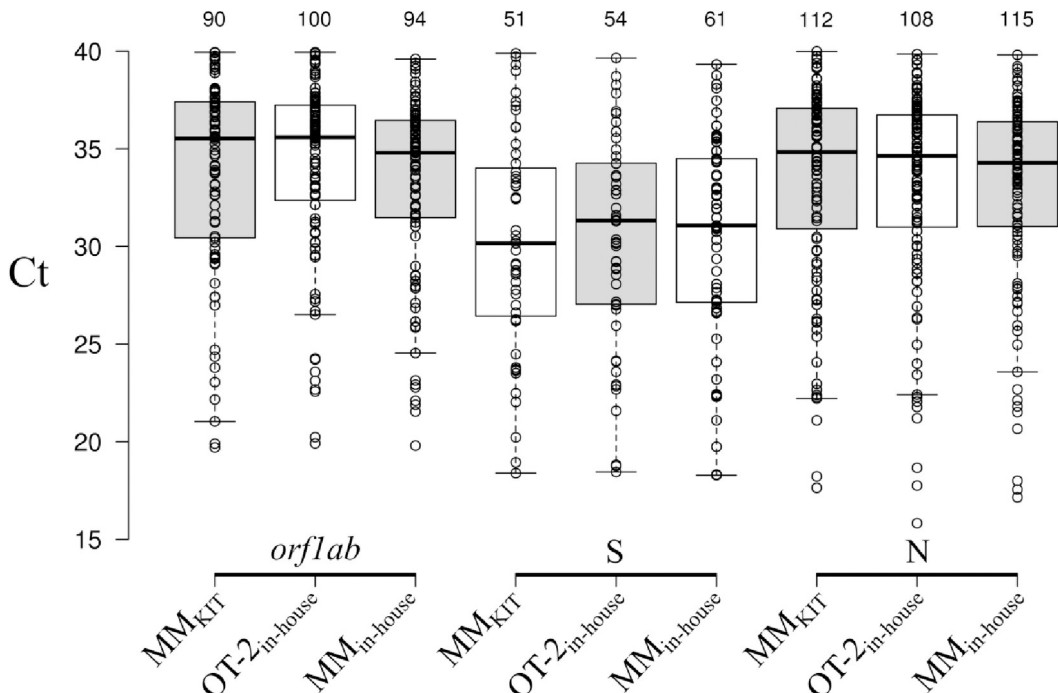

**Fig 2. Boxplot showing the distribution of the Ct values obtained for the *orf1ab*, S and N targets with the MM$_{kit}$, OT-2$_{in\text{-}house}$ and MM$_{in\text{-}house}$ methods.** The y-axis shows the amplification cycle (Ct). The number of data points is shown over each data set.

## Discussion

In this work, two methods are presented for low-cost viral RNA extraction and detection of SARS-CoV-2 in clinical samples with a performance comparable to that of a commercial kit. Although in-house protocols extracted the DNA control less efficiently than the commercial kit, in clinical samples the overall sensitivity was not affected, probably because in-house protocols use larger sample starting volumes that compensate the lower efficiency.

Setting up the OT-2 system for RNA extraction required intensive programming for a non-experienced team but provided a cost-effective alternative capable of extracting 48 samples in a single run. The scripts used in this work were uploaded to the open access software repository GitHub, where many protocols for these and similar applications can be found. This should help other workers to avoid or reduce the programming steps. The protocol demands little hands-on time and uses easily available reagents. In addition, the equipment requires a lower investment compared to other extraction systems, making it suitable for middle to low resource facilities. The MagMAX$^{TM}$ Express-96 is a fast, semi-automatic equipment capable of extracting 96 samples per run. The MM$_{kit}$ provides reagents for inactivation, lysis, washing and elution and can be used for different matrices and samples types at a competitive cost. On the other hand, the MM$_{in\text{-}house}$ protocol takes advantage of the semi-open approach of this system by substituting the commercial solutions by other chemicals making it more cost effective. In both cases, preparing the deepwell plates with buffers and samples must be done manually, increasing the total extraction time and the manipulation error risk. The main drawback of the two in-house protocols comes when handling viscous samples, the presence of mucus, highly viscous polysaccharides, leukocytes, erythrocytes, hemoglobin, proteases and cell detritus with high amounts of cellular nucleic acids can preclude the RNA extraction or inhibit the PCR

reaction [10–12]. To partially overcome this problem, samples can be heated and vortexed or subjected to a centrifugation prior to extraction, though this would increase the hands-on and response times. When using the $MM_{in\text{-}house}$, the eluted samples may have in some cases traces of the magnetic beads though this did not affect the performance of the RT-PCR reactions.

With the three systems negative controls were in all cases negative, indicating that cross-contamination is not an issue when working with primary samples in these systems. Nevertheless, same as with non-automated systems, precautions should be taken to keep pre-PCR and post-PCR operations separated. If an OT-2 module is used to set-up post-PCR reactions (eg. sequencing) it should not be used to extract nucleic acids from samples unless it is thoroughly decontaminated.

## Study strength and limitations

The major strength of the work is that the methods were tested with real samples in a clinical microbiology laboratory. The comparison between methods was not systematic, and this is an important limitation. In fact, the design was intended to optimize the results for each system, and so the inputs were different. A systematic exploration of the input sample volumes, as well as other reagent volumes would be helpful and informative, but it was beyond the aim of this work that was to compare the performance of the systems in a clinical laboratory environment with real samples.

## Conclusion

In summary, the two *in-house* nucleic acid extraction methods presented here are efficient for the high throughput diagnosis of SARS-CoV-2 at fraction of the costs of other commercial kits without losing sensitivity.

## Supporting information

**S1 Table. Cts obtained with the simulated samples at three different concentrations (Ctss) and Cts of the diluted controls used to calculate the efficiency (Ctpc).** (n.d. not detected). (TIF)

**S1 Fig. Bland-Altman plot showing the agreement between the three methods with three marker genes.** The same set of samples was extracted with the three methods and analyzed with a commercial PCR targeting three marker genes. The horizontal axes show the average Cts, and the vertical axes show the difference between Cts for each sample with the two methods indicated. The horizontal continuous line marks the average difference, and the discontinuous lines indicate the 95% limits of agreement (average difference ± 1.96 standard deviation of the difference). (TIF)

**S1 Data.** (XLSX)

## Acknowledgments

We are grateful to the COVID Warriors organization (www.covidwarriors.org) for donating the Opentrons OT-2 stations.

The authors also acknowledge the additional members of the SARS-CoV-2 Working Group: María Dolores Montero-Vega, María Pilar Romero, Silvia García-Bujalance, Emilio Cendejas Bueno, Carlos Toro-Rueda, Guillermo Ruiz-Carrascoso, Fernando Lázaro Perona,

Iker Falces-Romero, Almudena Gutierrez-Arroyo, Patricia Girón de Velasco-Sada, Mario Ruiz-Bastián, Marina Alguacil-Guillén, Patricia González-Donapetry, Gladys Virginia Guedez-López, Paloma García-Clemente, María Gracia Liras Hernandez, Consuelo García-Sanchez, Miguel Sánchez-Castellano and Sol San José-Villar. Servicio de Microbiología, Hospital Universitario La Paz, IdiPAZ, Paseo de La Castellana 261, 28046 Madrid, Spain.

## Author Contributions

**Conceptualization:** Fernando Lázaro-Perona, Jesús Mingorance, Julio García-Rodriguez.

**Data curation:** Fernando Lázaro-Perona, Almudena Gutiérrez-Arroyo, Jesús Mingorance.

**Formal analysis:** Fernando Lázaro-Perona, Jesús Mingorance.

**Investigation:** Fernando Lázaro-Perona, Marina Alguacil-Guillén, Almudena Gutiérrez-Arroyo.

**Methodology:** Fernando Lázaro-Perona, Carlos Rodriguez-Antolín, Marina Alguacil-Guillén, Almudena Gutiérrez-Arroyo.

**Project administration:** Jesús Mingorance, Julio García-Rodriguez.

**Resources:** Fernando Lázaro-Perona.

**Software:** Fernando Lázaro-Perona, Carlos Rodriguez-Antolín.

**Supervision:** Jesús Mingorance, Julio García-Rodriguez.

**Validation:** Julio García-Rodriguez.

**Writing – original draft:** Fernando Lázaro-Perona, Jesús Mingorance.

**Writing – review & editing:** Jesús Mingorance, Julio García-Rodriguez.

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
