## [Decision Letter · Decision Letter 0]

1 Dec 2020

PONE-D-20-34741

Evaluation of two automated low-cost RNA extraction protocols for SARS-CoV-2 detection

PLOS ONE

Dear Dr. Mingorance,

Thank you for submitting your manuscript to PLOS ONE. After careful consideration, we feel that it has merit but does not fully meet PLOS ONE’s publication criteria as it currently stands. Therefore, we invite you to submit a revised version of the manuscript that addresses the points raised during the review process.

As appended below, the reviewers have raised major concern/critique and suggested further justification/work to consolidate the findings. Do go through the comments and amend the MS accordingly. Furthermore

1- Avoid using "We, our". Use impersonal phrasing throughout the text

2- Full vendor details should include company, city (state), and country. Please amend and be consistent throughout the MS

3- Please use uL, mL, L throughout the text

4- The P letter for statistical value should be uppercase-italic face letter throughout the text

5- Please use min instead of minute throughout the text

6- What are the "Study strength and limitations". Please add this in a separate section headed as stated. It should be before conclusion section

7- Conclusion should be in a separate section headed as stated. What are the clinical relevance and future perspective. Please address them in this section as well

8- Abstract should be "structured one". i.e. it should include background, methods, Results and Conclusion

We look forward to receiving your revised manuscript.

Kind regards,

A. M. Abd El-Aty

Academic Editor

PLOS ONE

Journal Requirements:

2. Thank you for including your ethics statement:  "The work was done with anonymous discarded remnants of clinical samples set to be destroyed. No previous data on the patients or the samples themselves was collected. The results of the assays performed had no impact in any way on patient management.".   

4. One of the noted authors is a group or consortium [SARS-CoV-2 Working Group]. In addition to naming the author group, please list the individual authors and affiliations within this group in the acknowledgments section of your manuscript. Please also indicate clearly a lead author for this group along with a contact

Reviewers' comments:

Reviewer's Responses to Questions

**Comments to the Author**

1. Is the manuscript technically sound, and do the data support the conclusions?

Reviewer #1: Yes

Reviewer #2: Yes

Reviewer #3: Partly

Reviewer #4: Yes

2. Has the statistical analysis been performed appropriately and rigorously? 

Reviewer #1: Yes

Reviewer #2: Yes

Reviewer #3: Yes

Reviewer #4: Yes

3. Have the authors made all data underlying the findings in their manuscript fully available?

Reviewer #1: Yes

Reviewer #2: Yes

Reviewer #3: Yes

Reviewer #4: Yes

4. Is the manuscript presented in an intelligible fashion and written in standard English?

Reviewer #1: Yes

Reviewer #2: Yes

Reviewer #3: Yes

Reviewer #4: Yes

5. Review Comments to the Author

Reviewer #1: 1. More references should be added. For example, references related to the reagent choices used in this work should be added.

2. I would like to see more justifications in using different sample input and elution volume among the three methods. Adding more volume in the MM in house method, of course, can improve the total yield. An better approach is using standardized input and elution volume across the three methods.

3. The authors claimed lysis buffer was not used in the two in-house methods and there should be more discussions on why the traditional GTC lysis buffer is not needed but still was able to deliver good yield. However, it is probably has something to do with the samples were already mixed with GTC and heated before being used in the extraction process (Sample collection stage, lines 59-65). Therefore, it is incorrect to suggest that the in-house protocols lack efficient lysis steps.

4. Addition discussion on risk of cross-contamination on the OTC platform should be discussion. Did the authors use negative samples along with positive samples in the same run?

5. Overall, the results of the study is not unexpected. In-house reagents are expected to be cheaper than reagent kits bought from vendor. Also, method using large input can compensate extraction yield. The impact of this work is on the value of using a open-source, less expensive, device to perform sample preparation. I would like to see how much time was spent on programming the device before it was ready to collect data for this study. In addition, if the device is later used for master mix preparation, will false positive arise?

Reviewer #2: Lázaro-Perona et al. describe open-source, bead-based RNA extraction methods that are significantly less expensive than the standard MagMAX kit used for SARS-CoV-2 diagnostics. These methods perform comparably to the MagMAX kit when evaluated using a large collection of 141 SARS-CoV-2 positive clinical samples. The authors’ alternative RNA extraction methods could reduce unnecessary reliance on expensive robotic systems and proprietary RNA extraction reagents for SARS-CoV-2 testing, making this an extremely useful advance. I recommend that this work be published with minor revisions.

Major comments:

1. It would be helpful to provide a more detailed, step-by-step description of the different protocols used. Instead of saying that the manufacturer’s instructions were followed for the MagMAX kit, it would be worth listing the steps explicitly. The manufacturer’s protocol may change in the future, and so it is important to record the current protocol at the time that this work was performed. It was also a bit difficult to follow the description of the two in-house protocols. Perhaps it would be clearer to format each protocol as a numbered list of steps. The authors should provide the product information for the 96-well plates and 96-well magnet that they used. How long were the plates kept on the magnet to pull the beads to the side, and how long were the beads left to air dry after the second ethanol wash?

2. In the experiment described in the first Results section, the in-house methods showed lower fractional recovery of nucleic acid than the MagMAX protocol. Because these experiments were performed with short pieces of DNA rather than viral RNA, it is unclear whether these measurements are informative. If possible, it would be best to repeat these experiments using viral RNA (e.g., pooled leftover RNA from clinical samples).

3. Fig. 1 is a bit difficult to interpret at a glance, and it would help to have a summary panel showing the overall percentage of samples that tested positive using each protocol (i.e., 114/141 = 81%, 111/141 = 79%, and 118/141 = 84%).

Minor comments:

1. It should be stated in the methods what concentrations of guanidine isothiocyanate and carrier RNA were added to the samples.

2. Line 67: “two equipments” isn’t grammatically correct. Perhaps say “two pieces of equipment” or “two liquid handling robots”.

3. In the “Statistical analysis” section, they say that they used the Wilcoxon signed-rank test because the Ct data were not normally distributed, but then say in the next sentence that they used a paired t-test (which assumes normality). Please clarify.

4. Lines 191 and 203: I think they mean “investment” rather than “inversion”

5. They mention that the protocols “lack an efficient lysis step”, unlike the MagMAX kit. Did they include a proteinase K incubation step in the MagMAX protocol, and might adding this step to their protocol improve its performance?

Reviewer #3: The manuscript entitled “Evaluation of two automated low-cost RNA extraction protoocls for SARS-CoV-2 detection” submitted by Minorance et. al compares multiple extraction methods for COVID-19 testing including 2 automated lab developed protocols. In the manuscript the authors used a magnetic bead base extraction with the Opentrons OT-2 liquid robot and the MagMAX express system. Comparison of the assays was performed first by a dilution series of a positive control and later by testing of 141 SARS-CoV-2 positive patient samples. The QC testing demonstrated similar efficiencies between the in house protocols to the industry method. When testing of the 141 clinical samples 123 were positive by at least one extraction method and 18 were negative on all three methods. The authors suggested this is due to degradation of specimens. Finally, the authors compared overall cost and demonstrated cost effectiveness of the lab developed extractions due to reagent cost. Overall, the study was well written and adds to the data about methods for improving COVID-19 testing during the pandemic. Extractions are some of the most time consuming and limiting steps and use of more automation should help improve laboratory workflows. There are a few modifications needed:

Major Comments:

- The data for efficiency is very interesting, but it would be helpful to add a table demonstrating QC concentration followed by the number of tests that were detected with that extraction method *similar to LoD result.

- It is a bit concerning that there were so many FN results from the 141 samples tested. What were the storage conditions of the specimens, were they stored at 2-8C or frozen and how long. Was the standard of care data extracted using one of these extraction methods?

- What was the transport media used for the specimens and did this have any effect if it was multiple media types.

Minor Comments

- Ln 98-100 What was the concentration of the viral material used and to save needing to look it up, was this a capsulated RNA, inactivated virus, or free RNA?

- Add y-axis legend for Figures 1 and 2

Reviewer #4: Lázaro-Perona et al. have compared two automated RNA extraction protocols for SARS-CoV-2 qPCR detection. There are few aspects that should be clarified for the readers to enable them to repeat the protocol and utilize it.

There has been previous work on automated RNA extraction protocols for SARS-CoV-2 incl. those utilizing alternative reagents (doi:10.1261/rna.076232.120). It will be pertinent to briefly mention some or refer to reviews on same for readers to obtain a perspective of this work.

Please provide link to protocol for MagMAX CORE Nucleic Acid Purification kit (MMkit). Is the 500 uL sample volume compatible with protocol recommendations? For instance, is there anything in the MMkit protocol (volumes, steps, use of GTC or other finer details) that is different from protocol used in the paper.

Please give details of exact reagents used for replication. For instance, viral transport medium (specification/manufacturer etc. or constituents if lab made).

"MagMAX TM with the same script as the commercial kit and with the reagents used in the OT-2 method"- What is meant by same script? Same volumes, protocol. It might be good to describe this protocol & volumes at least in supporting info for clarity. For instance, it mentions "a final volume of 500 uL for the OT-2in-house and 1000 uL for the MMin-house"- Does different volumes mean different concentrations here? For equivalent comparison why was the volume increases for MMin-house esp. since MB volume used is same (40 uL)? Based on this your extraction efficiency will be different? Why is it 200 uL for MMkit if the protocol is same as MMin-house? The protocols have differences in use of Lysis/Binding

buffer & Isopropanol. Do these affect the efficiencies or explain why the comparison of efficiencies are valid if these details in protocols are different. Maybe explain in introduction that in-house protocols are "extraction free protocols" or direct sample addition protocols.

In line 100, "mixing 10 ul of the positive control with 490 ul of viral transport medium and 500 ul of GTC". So for the 3 different protocols do you take 200, 250 & 500 ul of sample and mix with corresponding volume ration of GTC? Or are these volumes only relevant for clinical samples? If the dilution is different is expected concentration of RNA different?

What is the starting conc. of the DNA positive control TaqMan 2019-nCoV Control Kit v1 ?

Line 103-"All mock samples were prepared in triplicate"-Is that n=3 for each platform or is it n=1 for each? i.e were 3 samples run on each platform or one. Please clarify this.

"1:100, 1:1000 and 1:10000 respectively."- Assuming this is volume percent?

"corrected by the dilution factor and the amount of initial sample used in each protocol (Ctpc) (R = 2-ΔCt=2-(Ctss-Ctpc))."- I am assuming this is key line which explains how different initial sample volumes are corrected for concentration? Maybe clarify it or elaborate the eqn. for better understanding.

Line 115-"positive control with the qPCR conditions recommended by the manufacturer." - Please cite the protocol if available online or if published.

Line 128- "Efficiencies of the extraction protocols"-Please provide data used to calculate the extraction efficiency in tabular or other form (maybe in supporting data if not relevant to main text).

Line 85-"the mixtures are incubated for 5 minutes at room temperature"-Is there any vortexing or mixing.

Line 118-What volumes were dispensed for each of the protocols?

Line 153-"probably due to sample degradation during the collection period"-Did the human RNaseP genes get detected in these -ve samples? What was the original test done to determine positive status of the collected 141 SARS-CoV-2 positive nasopharyngeal swabs?

"The 18 samples that showed discrepancies between methods had high Ct values"- What samples are being referred here? What is nature of discrepancies?

"method used in 43%, 49% and 49% of the samples"-Does it mean all of the 3 targets were detected? Please clarify.

"and was irrelevant for the diagnostic"-Please explain this statement

"Samples with multi-target amplification"-Please explain multi-target amplification (amplifying at least 2 or more of the genes?)

"probably because in-house protocols have a larger sample starting volume that"- Please explain this statement. Sample volumes will have different concentrations?

"the presence of mucus can preclude the RNA extraction or inhibit the PCR reaction"-Please cite a reference for this statement.

Line 27-"designed to be use with"- used with

Line 67-"we used two equipments"-noun equipment does not have a plural form

Line 77-"as ethanol absolute"- such as ethanol or correct the sentence

Line 190 "OT-2 protocol for 96 samples resulted 40€ more expensive than the MMin-house, the"-Please correct sentence

Line 191-"required initial inversion is significantly lower, being approximately 10.000€ for the"-Inversion might be wrong word. Is it 10,000€ & 50,000€.

"The protocol requires few hands-on time" & "requires a lower inversion"- Please correct sentence

6. PLOS authors have the option to publish the peer review history of their article (what does this mean?). If published, this will include your full peer review and any attached files.

Reviewer #1: No

Reviewer #2: No

Reviewer #3: No

Reviewer #4: No

---

## [Author Response · Author response to Decision Letter 0]

30 Dec 2020

Reviewer #1: 

More references should be added. For example, references related to the reagent choices used in this work should be added.

- Done as requested. References 5-8 added.

I would like to see more justifications in strengths and limitations in using different sample input and elution volume among the three methods. Adding more volume in the MM in house method, of course, can improve the total yield. An better approach is using standardized input and elution volume across the three methods.

- The reviewer is right and there is room for further optimization of the in house methods, but the aim of the work was not to do a systematic comparison of the systems, but to evaluate the utility of the in house systems for the clinical microbiology laboratory in an emergency situation in which there was shortage of commercial kits, so we used the maximal sample volume allowed by each system. A paragraph along this line has been included in the section " strengths and limitations".

The authors claimed lysis buffer was not used in the two in-house methods and there should be more discussions on why the traditional GTC lysis buffer is not needed but still was able to deliver good yield. However, it is probably has something to do with the samples were already mixed with GTC and heated before being used in the extraction process (Sample collection stage, lines 59-65). Therefore, it is incorrect to suggest that the in-house protocols lack efficient lysis steps.

- The reviewer is right. The text has been corrected.

Addition discussion on risk of cross-contamination on the OTC platform should be discussion. Did the authors use negative samples along with positive samples in the same run?

- Negative controls were included in all the runs. This information has been added to the text, as well as a paragraph on cross-contamination in the discussion.

I would like to see how much time was spent on programming the device before it was ready to collect data for this study. In addition, if the device is later used for master mix preparation, will false positive arise?

- Protocol programming took several days for a non-experienced team. The scripts were uploaded to the GitHub repository that contains an increasing number of open-access scripts to help workers to reduce or avoid programming. Once the programming is done, setting up the OT-2 module, even if some readjustments were needed, might take one or two days. This has been commented in the text.

There were no cross-contaminations when working with primary samples, nevertheless all the usual precautions taken when working with diagnostic PCR should be taken, so we routinely perform master mixes in a different module. Post-PCR operations (eg. NGS library preparation) should never be done in a module intended for pre-PCR operations. This has been commented in the text.

Reviewer #2: 

Major comments:

It would be helpful to provide a more detailed, step-by-step description of the different protocols used. Instead of saying that the manufacturer’s instructions were followed for the MagMAX kit, it would be worth listing the steps explicitly.

It was also a bit difficult to follow the description of the two in-house protocols. Perhaps it would be clearer to format each protocol as a numbered list of steps. The authors should provide the product information for the 96-well plates and 96-well magnet that they used. How long were the plates kept on the magnet to pull the beads to the side, and how long were the beads left to air dry after the second ethanol wash?

- The information requested has been added to the text.

In the experiment described in the first Results section, the in-house methods showed lower fractional recovery of nucleic acid than the MagMAX protocol. Because these experiments were performed with short pieces of DNA rather than viral RNA, it is unclear whether these measurements are informative. If possible, it would be best to repeat these experiments using viral RNA (e.g., pooled leftover RNA from clinical samples).

- The reviewer is right, but as far as we know (communication from the company) the control DNA is a plasmid containing a synthetic construction with all the target sequences. They are not short pieces. The suggestion of the reviewer of doing the comparison with viral RNA is interesting and was discussed by us, but after seeing the results obtained with the samples we felt that it was not necessary. As commented to reviewer#1 the aim of the work was not to do a systematic comparison of the systems, but to evaluate the utility of the in house systems for the clinical microbiology laboratory in an emergency situation in which there was limited availability of commercial kits.

Fig. 1 is a bit difficult to interpret at a glance, and it would help to have a summary panel showing the overall percentage of samples that tested positive using each protocol (i.e., 114/141 = 81%, 111/141 = 79%, and 118/141 = 84%). 

- This information has been added to the figure legend in order to make it more clear.

1. It should be stated in the methods what concentrations of guanidine isothiocyanate and carrier RNA were added to the samples.

- Done as requested.

2. Line 67: “two equipments” isn’t grammatically correct. Perhaps say “two pieces of equipment” or “two liquid handling robots”.

- Corrected

3. In the “Statistical analysis” section, they say that they used the Wilcoxon signed-rank test because the Ct data were not normally distributed, but then say in the next sentence that they used a paired t-test (which assumes normality). Please clarify.

- This was an error. Reference to the paired t-test has been deleted.

Lines 191 and 203: I think they mean “investment” rather than “inversion”

- Corrected

They mention that the protocols “lack an efficient lysis step”, unlike the MagMAX kit. Did they include a proteinase K incubation step in the MagMAX protocol, and might adding this step to their protocol improve its performance?

- As pointed by reviewer #1 the statement was not correct. The text has been corrected. Proteinase K was included only in the MMkit method.

Reviewer #3: 

The data for efficiency is very interesting, but it would be helpful to add a table demonstrating QC concentration followed by the number of tests that were detected with that extraction method *similar to LoD result.

- A supplementary table has been added.

It is a bit concerning that there were so many FN results from the 141 samples tested. What were the storage conditions of the specimens, were they stored at 2-8C or frozen and how long. Was the standard of care data extracted using one of these extraction methods?

- The standard of care was extracted with the MMkit and the PCR was the same, but the samples were processed immediately after inactivation with guanidinium chloride. The remaining inactivated material was stored at 4ºC for 24-48 hours until all the samples for this work had been collected. During this storage at 4ºC viral RNA may be lost in up to 30% of the samples. We have observed this phenomenon several times and have added a reference from another group documenting similar observations (ref. 9).

What was the transport media used for the specimens and did this have any effect if it was multiple media types.

- Transport medium and swab were part of a kit commercialized by Deltalab. The reference to Deltalab that was after "swab" has been moved to the end of the sentence to make this more clear. The company does not disclose the composition of the medium, just that it contains bacterial and fungal growth inhibitors. During the collection period a single transport media was used.

- Ln 98-100 What was the concentration of the viral material used and to save needing to look it up, was this a capsulated RNA, inactivated virus, or free RNA?

- As stated in the text the control DNA was plasmid. The clinical samples were real samples taken from patients and containing intact virus.

- Add y-axis legend for Figures 1 and 2

- Corrected

Reviewer #4: 

There has been previous work on automated RNA extraction protocols for SARS-CoV-2 incl. those utilizing alternative reagents (doi:10.1261/rna.076232.120). It will be pertinent to briefly mention some or refer to reviews on same for readers to obtain a perspective of this work.

- Several references have been added.

Please provide link to protocol for MagMAX CORE Nucleic Acid Purification kit (MMkit). Is the 500 uL sample volume compatible with protocol recommendations? For instance, is there anything in the MMkit protocol (volumes, steps, use of GTC or other finer details) that is different from protocol used in the paper.

- The protocol has been described and a reference to the original pdf has been included. The MMkit is the same as the one described by the company.

Please give details of exact reagents used for replication. For instance, viral transport medium (specification/manufacturer etc. or constituents if lab made).

- As stated above, viral transport medium and swab were part of a kit commercialized by Deltalab. The reference to Deltalab that was after "swab" has been moved to the end of the sentence to make this more clear. The company does not disclose the composition of the medium, just that it contains bacterial and fungal growth inhibitors. During the collection period this was the only transport media was used.

"MagMAX TM with the same script as the commercial kit and with the reagents used in the OT-2 method"- What is meant by same script? Same volumes, protocol.

- The protocols have been included in the text to make it more clear. The same script means the same instrument program.

It might be good to describe this protocol & volumes at least in supporting info for clarity. For instance, it mentions "a final volume of 500 uL for the OT-2in-house and 1000 uL for the MMin-house"- Does different volumes mean different concentrations here? 

- The protocols have been included in the text to make it more clear. Different volume means different starting volume. The samples were the same, so the starting concentrations were the same. This is discussed now in the strengths and limitations section.

For equivalent comparison why was the volume increases for MMin-house esp. since MB volume used is same (40 uL)? Based on this your extraction efficiency will be different? 

- Increasing the sample volume increases the amount of material to be extracted and therefore the final yield. MBs are far from saturation and have capacity to bind higher amounts of RNA, we do not think that extraction efficiency is different, but the final yield is higher.

Why is it 200 uL for MMkit if the protocol is same as MMin-house? The protocols have differences in use of Lysis/Binding buffer & Isopropanol.

- The reviewer is right, the protocols are different and start with different amounts of sample. The aim of the work was not as much to do a systematic comparison of the systems as to evaluate the utility of the in house systems for the clinical microbiology laboratory, so we used the maximal sample volume possible in each system. A paragraph along this line has been included in the section " strengths and limitations".

Do these affect the efficiencies or explain why the comparison of efficiencies are valid if these details in protocols are different. Maybe explain in introduction that in-house protocols are "extraction free protocols" or direct sample addition protocols.

- The protocols were different, as stated above. Comparison of the efficiencies was intended to be used as a guideline.

In line 100, "mixing 10 ul of the positive control with 490 ul of viral transport medium and 500 ul of GTC". So for the 3 different protocols do you take 200, 250 & 500 ul of sample and mix with corresponding volume ration of GTC? Or are these volumes only relevant for clinical samples?

- The 200, 250 & 500 ul already include GTC.

If the dilution is different is expected concentration of RNA different?

- Dilution is the same, 1:1. As explained above, the starting amount is different.

What is the starting conc. of the DNA positive control TaqMan 2019-nCoV Control Kit v1 ?

- 1 x 104 copies/µL. This information has been included in the text.

Line 103-"All mock samples were prepared in triplicate"-Is that n=3 for each platform or is it n=1 for each? i.e were 3 samples run on each platform or one. Please clarify this.

- It was one triplicate per dilution per platform.

"1:100, 1:1000 and 1:10000 respectively."- Assuming this is volume percent? 

- This is dilution. The text has been rewritten to clarify.

"corrected by the dilution factor and the amount of initial sample used in each protocol (Ctpc) (R = 2-ΔCt=2-(Ctss-Ctpc))."- I am assuming this is key line which explains how different initial sample volumes are corrected for concentration? Maybe clarify it or elaborate the eqn. for better understanding.

- The explanation was not clear. The positive controls were diluted to match the amounts introduced in the extraction in each protocol, so no further corrections are needed. The sentence has been rewritten to make it more clear.

Line 115-"positive control with the qPCR conditions recommended by the manufacturer." - Please cite the protocol if available online or if published.

- Done as suggested.

Line 128- "Efficiencies of the extraction protocols"-Please provide data used to calculate the extraction efficiency in tabular or other form (maybe in supporting data if not relevant to main text).

- A supplementary table has been added.

Line 85-"the mixtures are incubated for 5 minutes at room temperature"-Is there any vortexing or mixing.

- The machines mix by repeated pippeting. The protocols have been included in the manuscript and this is explicitly stated.

Line 118-What volumes were dispensed for each of the protocols?

- To make it more clear the protocols have been included in the manuscript and the amounts are explicitly stated.

Line 153-"probably due to sample degradation during the collection period"-Did the human RNaseP genes get detected in these -ve samples? What was the original test done to determine positive status of the collected 141 SARS-CoV-2 positive nasopharyngeal swabs?

- The standard of care was extracted with the MMkit and the PCR was the same, but the samples were processed immediately after inactivation with guanidinium chloride. The inactivated material left after taking the sample for the standard of care was stored at 4ºC for 24-48 hours until the 141 samples were collected and used. During this storage at 4ºC viral RNA may be lost in up to 30% of the samples while the RNAseP is still positive. We have observed this phenomenon several times and have added a reference from another group documenting similar observations (ref. 9).

"The 18 samples that showed discrepancies between methods had high Ct values"- What samples are being referred here? What is nature of discrepancies?

- 18 samples were negative by one of the three methods. The sentence has been rewritten to make it more clear.

"Method used in 43%, 49% and 49% of the samples"-Does it mean all of the 3 targets were detected? Please clarify.

- The sentence has been rewritten to make it more clear.

"and was irrelevant for the diagnostic"-Please explain this statement

- The sentence has been rewritten to make it more clear.

"Samples with multi-target amplification"-Please explain multi-target amplification (amplifying at least 2 or more of the genes?) 

- The sentence has been rewritten to make it more clear.

"probably because in-house protocols have a larger sample starting volume that"- Please explain this statement. Sample volumes will have different concentrations?

- The volumes are different and the concentrations are the same, therefore the starting amount of RNA would be different.

"the presence of mucus can preclude the RNA extraction or inhibit the PCR reaction"-Please cite a reference for this statement.

- Three references have been added (10-12).

Line 27-"designed to be use with"- used with 

- Done

Line 67-"we used two equipments"-noun equipment does not have a plural form

- Done

Line 77-"as ethanol absolute"- such as ethanol or correct the sentence 

- Done

Line 190 "OT-2 protocol for 96 samples resulted 40€ more expensive than the MMin-house, the"-Please correct sentence 

- Done

Line 191-"required initial inversion is significantly lower, being approximately 10.000€ for the"-Inversion might be wrong word. Is it 10,000€ & 50,000€. 

- Done

"The protocol requires few hands-on time" & "requires a lower inversion"- Please correct sentence

- Done

---

## [Decision Letter · Decision Letter 1]

18 Jan 2021

Evaluation of two automated low-cost RNA extraction protocols for SARS-CoV-2 detection

PONE-D-20-34741R1

Dear Dr. Mingorance,

We’re pleased to inform you that your manuscript has been judged scientifically suitable for publication and will be formally accepted for publication once it meets all outstanding technical requirements.

Kind regards,

A. M. Abd El-Aty

Academic Editor

PLOS ONE

Reviewers' comments:

Reviewer's Responses to Questions

**Comments to the Author**

1. If the authors have adequately addressed your comments raised in a previous round of review and you feel that this manuscript is now acceptable for publication, you may indicate that here to bypass the “Comments to the Author” section, enter your conflict of interest statement in the “Confidential to Editor” section, and submit your "Accept" recommendation.

Reviewer #1: All comments have been addressed

Reviewer #2: All comments have been addressed

Reviewer #3: All comments have been addressed

Reviewer #4: All comments have been addressed

2. Is the manuscript technically sound, and do the data support the conclusions?

Reviewer #1: Yes

Reviewer #2: Yes

Reviewer #3: Yes

Reviewer #4: Yes

3. Has the statistical analysis been performed appropriately and rigorously? 

Reviewer #1: Yes

Reviewer #2: Yes

Reviewer #3: Yes

Reviewer #4: Yes

4. Have the authors made all data underlying the findings in their manuscript fully available?

Reviewer #1: Yes

Reviewer #2: Yes

Reviewer #3: Yes

Reviewer #4: Yes

5. Is the manuscript presented in an intelligible fashion and written in standard English?

Reviewer #1: Yes

Reviewer #2: Yes

Reviewer #3: Yes

Reviewer #4: Yes

6. Review Comments to the Author

Reviewer #1: (No Response)

Reviewer #2: (No Response)

Reviewer #3: The authors have answered each of my comments and the comments from the other authors. I have no additional suggestions for the author.

Reviewer #4: All the review comments and detailed protocol descriptions have been well incorporated. The change incorporated will enable readers to reproduce the results described in the protocol.

7. PLOS authors have the option to publish the peer review history of their article (what does this mean?). If published, this will include your full peer review and any attached files.

Reviewer #1: No

Reviewer #2: No

Reviewer #3: No

Reviewer #4: **Yes: **Harikrishnan Jayamohan

---

## [Editor Report · Acceptance letter]

3 Feb 2021

PONE-D-20-34741R1 

Evaluation of two automated low-cost RNA extraction protocols for SARS-CoV-2 detection 

Dear Dr. Mingorance:

I'm pleased to inform you that your manuscript has been deemed suitable for publication in PLOS ONE. Congratulations! Your manuscript is now with our production department. 

Kind regards, 

on behalf of

Prof. A. M. Abd El-Aty 

Academic Editor

PLOS ONE